# Reduced Stroke Risk among Patients with Atrial Fibrillation Receiving Chinese Herbal Medicines Treatment: Analysis of Domestic Data in Taiwan

**DOI:** 10.3390/medicina56060282

**Published:** 2020-06-09

**Authors:** Li-Cheng Zheng, Hanoch Livneh, Wei-Jen Chen, Miao-Chiu Lin, Ming-Chi Lu, Chia-Chou Yeh, Tzung-Yi Tsai

**Affiliations:** 1Department of Traditional Chinese Medicine, Kaohsiung-Veterans General Hospital, Kaohsiung 81362, Taiwan; u9830035@cmu.edu.tw; 2Rehabilitation Counseling Program, Portland State University, Portland, OR 97207-0751, USA; livnehh@pdx.edu; 3Department of Chinese Medicine, Dalin Tzuchi Hospital, Buddhist Tzuchi Medical Foundation, 2 Minsheng Road, Dalin Township, Chiayi 62247, Taiwan; tough2915@hotmail.com; 4Department of Nursing, Dalin Tzuchi Hospital, Buddhist Tzuchi Medical Foundation, 2 Minsheng Road, Dalin Township, Chiayi 62247, Taiwan; df729376@tzuchi.com.tw; 5Division of Allergy, Immunology and Rheumatology, Dalin Tzuchi Hospital, The Buddhist Tzuchi Medical Foundation, 2 Minsheng Road, Dalin Township, Chiayi 62247, Taiwan; dm252940@tzuchi.com.tw; 6School of Medicine, Tzu Chi University, 701 Jhongyang Road Section 3, Hualien 97004, Taiwan; 7School of Post-Baccalaureate Chinese Medicine, Tzu Chi University, 701 Jhongyang Road Section 3, Hualien 97004, Taiwan; 8School of Chinese Medicine, China Medical University, Taichung 40402, Taiwan; 9Department of Environmental and Occupational Health, College of Medicine, National Cheng Kung University, 138 Sheng-Li Road, Tainan 70428, Taiwan; 10Department of Medical Research, Dalin Tzuchi Hospital, The Buddhist Tzuchi Medical Foundation, 2 Minsheng Road, Dalin Township, Chiayi 62247, Taiwan; 11Department of Nursing, Tzu Chi University of Science and Technology, 880 Chien-Kuo Road Section 2, Hualien 97004, Taiwan

**Keywords:** Chinese herbal medicines, atrial fibrillation, stroke, cohort study

## Abstract

*Background and objectives:* Patients with atrial fibrillation (AF) reportedly have a much higher risk of death due to stroke. Faced with this heavy burden, it remains unclear if the Chinese herbal medicines (CHMs), the most common form complementary and alternative medicine, can lower the risk of stroke for them. This study aimed to evaluate the association of CHMs use with stroke risk among them. *Materials and Methods:* From a nationwide database, 11,456 AF patients aged ≧ 20 years between 1998 and 2007 were identified. Afterwards, we enrolled 2670 CHMs users and randomly selected 2670 non-CHMs users using the propensity score method. The occurrence of stroke was recorded until the end of 2012. *Results:* Within the follow-up period, 671 CHMs users and 900 non-CHMs users developed stroke, with incidence rates of 33.02 and 45.46 per 1000 person-years, respectively. CHMs use was associated with a 30% lower stroke risk, especially for those receiving CHMs for over two years. *Conclusions:* The findings of the present study suggest that adding CHMs to conventional therapy could decrease subsequent stroke risk for AF patients. It is also suggested that prospective randomized trials are needed to further clarify if the detected association revealed in this study supports a causal link, and to identify the specific CHMs that may be beneficial to AF patients.

## 1. Introduction

Atrial fibrillation (AF) is a common irregular cardiac rhythm disorder [1], which has been proven to exacerbate the risk for several cardiac and cerebral vascular diseases, especially strokes [2,3]. A recent report further indicated that about 15% of stroke events in the United States are attributable to AF [4]. Once AF patients suffer from concomitant stroke, their chance of dying doubles compared to those who do not have a stroke [5]. Though warfarin is the commonly used oral anticoagulant for reducing the incidence of thromboembolism for AF patients [6,7], the prescription of warfarin has to be cautious, as the dosage is usually influenced by foods and other drugs, thus provoking the risk of bleeding and stroke [8]. As such, investigating the alternative treatments with fewer side effects is of great significance for disease management towards AF. 

Recently, Chinese herbal medicines (CHMs) have been used often to improve well-being for patients with chronic diseases, especially in patients with AF [9]. Based on traditional Chinese medicine theory, the blood-invigorating and stasis-removing (BISR) herbal products are commonly used to prevent or treat stroke [10]. These medications include Dan Shen Decoction, Bu Yang Huan Wu Tang, and Shu Jing Huo Xue Tang, all of which may slow the heart rate or inhibit the platelet aggregation, and as such, enhance cardiovascular health [9,11]. 

Though these medications have attracted attention for some time, studies evaluating their efficacy have still yielded inconsistent findings. Some recent studies showed that CHMs combined with conventional therapy may contribute to positive benefits in the management of stroke [9,12]. In contrast, a recent review of 54 studies indicated that the integration of CHMs and warfarin therapy would increase the risk of bleeding due to the exacerbation of the anticoagulant effect [13], implying that the combined use of CHMs and warfarin should be allocated much attention for AF patients. 

Following a detailed literature review, we observe that no large-scale studies have been conducted to adequately examine the influence of concomitant use of CHMs on the following stroke risk, especially in AF patients. Thus, this population-based study aimed to compare the risk of stroke among AF patients who received CHMs together with warfarin with those patients who received warfarin treatment only. 

## 2. Materials and Methods

### 2.1. Data Source

For this study, we used a publicly released cohort dataset from the Longitudinal Health Insurance Database (LHID). The LHID, a sub-dataset of the National Health Insurance (NHI) program made up of one million randomly sampled people who were alive in 2000, is a collection of all medical records of these individuals from 1996 to 2012. Constructed using a multistage stratified systematic sampling method, it is a representative sample, in both sex and age, of these one million insured individuals and the general population of Taiwan [14]. This database contains all NHI enrollment files, claims data and a prescription drugs registry, and provides comprehensive utilization information for the persons covered by the NHI. The current study was conducted in accordance with the Helsinki Declaration and was approved by the Institutional Review Board and Ethics Committee of Buddhist Dalin Tzu Chi Hospital, Taiwan (No. B10004021-2) on 31 August 2015, which waived the requirement for informed consent.

### 2.2. Study Subjects and Variables

In this work, to be designated as having a certain disease, the subject had to have a corresponding code by the International Classification of Disease, Ninth Revision, Clinical Modification (ICD-9-CM) in the diagnosis field. We recruited the subjects ≥ 20 years of age who sought ambulatory health care services due to AF (ICD-9-CM code: 427.31 or 427.89) in 1998 and 2007. Cases were classified as being AF if they had at least one admission code for AF, or three or more outpatient visits codes during the study period including the prescription of warfarin medication for a period of three months (*n* = 14,636). To establish a temporal link between AF and stroke, we omitted those who were followed for less than one year after the onset of AF (*n* = 157), and those with a diagnosis of stroke prior to the first AF onset (*n* = 3023). The stroke event was identified by the catastrophic illness registry. Only those patients with catastrophic illness certification due to stroke (ICD-9-CM code 430-438) were recruited. In Taiwan, insured residents with major diseases (e.g., cancer, autoimmune diseases, chronic mental diseases end-stage renal failure) can apply for a catastrophic illness certificate for exemption from co-payment. After this filtering process, a total of 11,456 AF subjects were identified. 

In Taiwan, only certified Chinese medicine physicians are entitled to prescribe CHMs. Accordingly, we used the drug days of Chinese herbal products to verify the CHM exposure of each AF patient. Among them, those who received CHMs for more than 30 days were considered CHM users, whereas those treated for 30 days or less were considered non-CHM users [15]. Based on this procedure, 8727 cases were designated as CHM users. A comparison cohort was randomly selected from the remaining insured AF patients who were not CHM users. For each patient receiving CHM treatment, a patient with no CHM treatment was selected, using 1:1 matching based on a propensity score. The propensity score was calculated using logistic regression on the basis of patient demographics and baseline comorbidities at enrollment. Ultimately, an equal number of subjects were enrolled in the CHM and non-CHM groups (Figure 1). The index date for AF subjects who were classified as non-CHM users was defined as the date of the first AF diagnosis, whereas the index date for AF cases who received CHMs was defined as the first date of receiving CHMs. The end date of the follow-up period for both groups was defined as the earliest of a diagnosis of stroke, withdrawal from the insurance program, or the date of December 31, 2012. The incidence rate of stroke was defined as number of events divided by the person-time at risk.

### 2.3. Covariate Assessment

Sociodemographic factors in this study included patient age, gender, income (for estimating insurance payments), and urbanization level of each subject’s residential area. The subjects’ monthly incomes were stratified into three levels: New Taiwan Dollars (NTDs) ≤ 17,880; 17,881–43,900; and ≥ 43,901. Urbanization levels were divided into urban (levels 1–2), suburban (levels 3–4) and rural (levels 5–7) areas. Level 1 refers to the “most urbanized” communities and level 7 refers to the “least urbanized” communities [16]. Baseline comorbidities for each subject included hypertension (ICD-9-CM 401–405), diabetes (ICD-9-CM 250), ischemic heart disease (ICD-9-CM 410–414, 443.89, 443.9 and 444), hyperlipidemia (ICD-9-CM 272), and congestive heart failure (ICD-9-CM 428, 402.01, 402.11, 402.91, 404.01, 404.11 and 404.91). These comorbidities were based on data in the medical records one year prior to the diagnosis of AF.

### 2.4. Statistical Analysis 

We employed the χ^2^ test and Student’s *t*-test to examine the baseline differences in sociodemographic characteristics and comorbidities between subjects with and without receiving CHMs. The incidence rate of stroke was presented with the number of cases per 1000 person-years (PYs). The Cox proportional hazards regression analysis was then applied to compute the hazard ratio (HR) with 95% confidence interval (CI) of stroke risk in association with CHM usage. To robustly explore the association of CHM therapy and the risk of stroke, we divided the CHM users into three subgroups: those with used CHMs for 30–365 days, those who used CHMs for 366–730 days and those who used CHMs for more than 730 days. Kaplan–Meier failure estimates of stroke risk were plotted and the differences between the three groups were examined using the log-rank test. The proportional-hazards assumption was verified using plots of log (-log (survival function) vs. log (time), and Schoenfeld residuals vs. time. All analyses were conducted using SAS version 9.3 (SAS Institute Inc., Cary, NC, USA). Values of *p* < 0.05 were considered to indicate statistical significance.

## 3. Results

The CHM user and non-CHM user cohorts were each comprised of 2670 patients. Table 1 shows the pertinent characteristics of the two groups, including distributions of age, sex, monthly income, residential area and comorbidities, indicating that the two groups were comparable in all characteristics.

Among the sampled AF patients, 1571 first episodes of stroke were identified; 900 were among the non-CHM users and 671 among the CHM users during the follow-up periods of 19,799.80 and 20,320.77 person-years (PYs), respectively. The incidence of stroke was lower among CHM users than among non-CHM users (33.02 vs. 45.46, respectively, per 1000 PYs), with an adjusted HR (95% CI) of 0.70 (0.63–0.77) (Table 2). Of note, those who used CHMs for more than 730 days had a 68% decreased risk of stroke (95% CI: 0.20–0.51) versus non-users. Based on the Kaplan–Meier survival curve and log-rank test results, the data also supported a statistically significant difference in the survival rate free from stroke across the three groups of users during the follow-up period. Those receiving CHMs for more than two years had a significantly lower incidence rate of stroke than those not receiving CHMs (*p* < 0.001) (Figure 2). 

The most commonly prescribed herbal products for AF patients were reported in Table 3. Among them, with the exception of Bu Yang Huan Wu Tang (BYHWT) and Fu Fang Dan Shen Pian (FFDSP), the other herbal products were found to be related to a significantly lower stroke risk.

## 4. Discussion

This is the first large nationwide population-based cohort study to compare the stroke risk in AF patients who were prescribed warfarin with or without the use of CHMs. This work yielded two major findings. First, the stroke risk was significantly lower in AF patients who received CHMs as compared to those who did not receive CHMs, especially for those receiving CHM treatment for more than two years. Second, the commonly prescribed herbal products, such as Shu Jing Hwo Shiee Tang (SJHST), Tong Qiao Huo Xue Tang (TQHXT), Xue Fu Zhu Yu Tang (XFZYT), Ge Xia Zhu Yu Tang (GXZYT), Fu Yuan Huo Xue Tang (FYHXT), Tao He Cheng Qi Tang (THCQT), Tao Hong Si Wu Tang (THSWT) and Shen Tong Zhu Yu Tang (STZYT), were found to be associated with a decreased risk of stroke.

Results of this long-term follow-up study found that AF patients who received CHMs exhibited a 30% decreased risk of stroke as compared to those who did not receive CHMs, echoing prior studies [9,12]. It could be inferred that the commonly used CHMs such as Danshen dripping pills or XFZYT may lessen the risk of stroke by the inhibition of platelet aggregation, free radical scavenging, and anti-inflammatory and neuroprotective properties. Nevertheless, our findings are inconsistent with a former report, which found that the concurrent use of CHMs may increase the bleeding risk for AF patients taking warfarin [17]. These conflicting findings may be attributed to methodological differences between the two studies. Saw and colleagues used a self-administered questionnaire to obtain information regarding the use of CHMs, but the method used may be influenced by recall bias [17]. Furthermore, the cross-sectional design employed in their study does not allow for any inference about any cause-and-effect relationships between the tested variables.

An additional contribution of this cohort study was the list of herb products that were related to the lower risk of stroke. For example, we noted that the use of THSWT was related to decreased stroke risk. A previous marine model indicated that this formula could reduce the risk of cerebral ischemia via suppressing caspase-3 and inducible nitric oxide synthase expression [18]. Additionally, the other commonly used CHMs for treating AF, such as TQHXT, XFZYT, GXZYT, and STZYT, were found to be associated with a lower risk of stroke, echoing prior studies [11,19]. Among these herbal formulae, Dang Gui (*Angelica sinensis Oliv. Diels*), Chuan Xiong (*Ligusticum striatum DC.* stem and root), Tao Ren (*Prunus persica L. Batsch*), and Hong Hua (*Carthamus tinctorius* L. flower) are the major ingredients that were proven to have BISR effects via the inhabitation of platelet aggregation and inflammatory responses [11,20]. In practice, the use of these formulae depends on the location of stasis. The prescription of TQHXT focuses on the area above the neck [21], and XFZYT focuses on the chest [22,23]. As for GXZYT, it is commonly used to treat the stasis occurring between the diaphragm and lower abdomen [24], and STZYT is utilized to remove the stasis on the limbs and musculoskeletal system [25].

This study also revealed that the use of FYHXT and THCQT was associated with a reduced risk of stroke. Tao Ren and Da Huang (*Rheum rhabarbarum* L. stalks) are the major ingredients of these formulae, and both have been proven to have anti-blood stasis effects [11]. In addition, SJHST can invigorate the blood, relax the blood vessels, and treat blood stagnation resulting from wind-cold-dampness, thereby decreasing the risk of cerebrovascular disease [10,26,27]. Alternatively, the failure to observe any link between the use of FFDSP and BYHWT and stroke risk may be attributed to the constituent differences. In practice, though these two formulae have much higher BISR influence, they were classically used to treat a variety of vascular diseases via their stasis-resolving and blood-activating influences, especially in the clinical manifestations after stroke onset [10,11].

Findings from the current study have important clinical and research implications. However, several limitations should be noted when interpreting these results. First, coding errors are always a possibility in an administrative database. To minimize this bias, we enrolled the subjects with the consistent diagnoses for at least three outpatient visits or at least one inpatient admission within the studying period. Meanwhile, it should be also remarked that the coding approach and data availability were similar between the two groups, and this misclassification bias could have likely been nondifferential and toward the null hypothesis, thus possibly inducing the underestimation rather than the overestimation of the observed estimate. Second, the LHID database does not include detailed information regarding the predictors of stroke, especially smoking. To this end, we performed sex-stratified analysis, and it supported an appreciably decreased risk of stroke for AF patients receiving CHMs, regardless of gender. Third, although our study revealed a substantial beneficial effect of CHM use on the reduction of stroke risk among AF patients, it must be recognized that participants were not initially randomly categorized into users and nonusers, and were only recruited from a single country. Therefore, caution should be exerted when interpreting the findings, especially the efficacy attributed to Chinese herbal products. Randomized controlled trials, encompassing additional countries, are recommended to corroborate the present findings, as well as to uncover the mechanisms underlying their successful application. These limitations notwithstanding, this study also had considerable advantages. The database applied in this study is representative of the entire Taiwanese population, and this large sample size ensured the obtaining of robust findings. In addition, the retrospective 15-year cohort study allowed us to methodically explore the relation between CHM use and stroke in AF persons. These findings may serve as a useful reference for future studies on this topic in other populations.

## 5. Conclusions

This study explored the possibility that patients with AF who receive CHMs have a reduced subsequent risk of stroke. Although beneficial effects of CHMs on reducing stroke risk were revealed in this study, future prospective randomized trials that overcome the limitations of this study are needed to provide more conclusive evidence of the association suggested by the findings of this study.

## Figures and Tables

**Figure 1 medicina-56-00282-f001:**
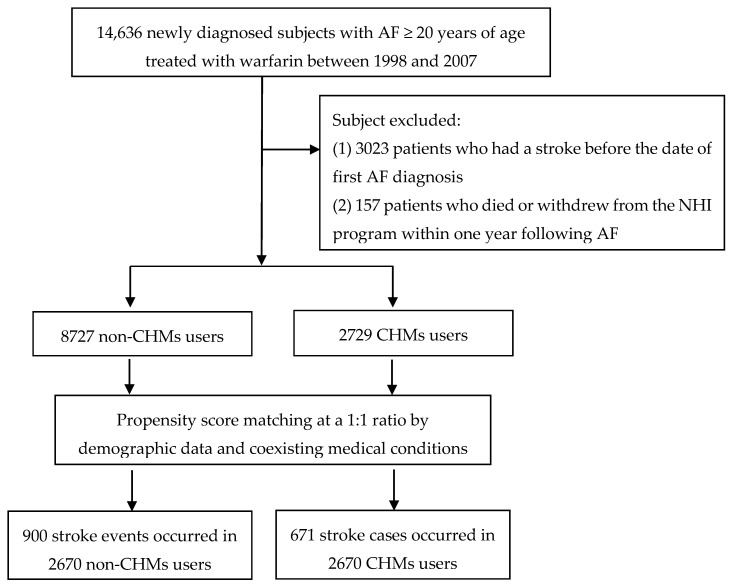
Flow chart of subject inclusion. AF: Atrial Fibrillation, CHM: Chinese herbal medicines.

**Figure 2 medicina-56-00282-f002:**
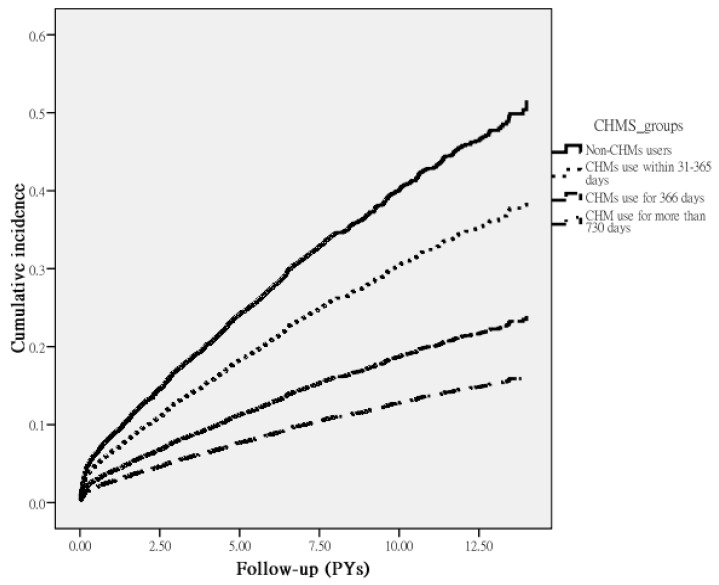
Cumulative incidence of stroke in AF patients with and without receiving CHM treatment during the study period (Log-rank test, *p* < 0.001).

**Table 1 medicina-56-00282-t001:** Subject demographic data and comorbidities.

Variables	CHMs Non-Users	CHMs Users	*p*
	*n* = 2670 (%)	*n* = 2670 (%)	
Age			0.32
Mean (standard deviation)	58.03 (15.88)	58.46 (15.18)	
Sex			0.87
Female	1377 (48.4)	1371 (51.3)	
Male	1293 (51.6)	1299 (48.7)	
Monthly income			0.38
Low	1193 (44.7)	1176 (44.0)	
Median	1386 (51.9)	1384 (51.8)	
High	91 (3.4)	110 (4.1)	
Residential area			0.27
Urban	1372 (51.4)	1351 (50.6)	
Suburban	404 (15.1)	447 (16.7)	
Rural	894 (33.5)	872 (32.7)	
Comorbidity			
Hypertension	1228 (46.0)	1208 (45.2)	0.58
Diabetes mellitus	372 (13.9)	405 (15.2)	0.20
Hyperlipidemia	364 (13.6)	342 (12.8)	0.37
Ischemic heart disease	959 (35.9)	990 (37.1)	0.38
Congestive heart failure	48 (1.8)	51 (1.9)	0.76

**Table 2 medicina-56-00282-t002:** Risk of stroke for AF patients with and without CHM use.

	Event	PYs	Incidence	Crude HR (95% CI)	Adjusted HR * (95% CI)
Non-CHMs users	900	19,799.80	45.46	1.00	1
CHMs users	671	20,320.77	33.02	0.73 (0.65–0.78)	0.70 (0.63–0.77)
CHMs use within 31–365 days	607	16,628.21	36.50423	0.78 (0.71–0.86)	0.77 (0.68–0.84)
CHMs use for 366–730 days	46	2353.39	19.54627	0.43 (0.32–0.57)	0.47 (0.35–0.63)
CHMs use for more than 730 days	18	1339.16	13.44126	0.30 (0.19–0.48)	0.32 (0.20–0.51)

Incidence rate is per 1000 person-years.; * Model adjusted for age, sex, urbanization level, monthly income, and comorbidities. HR: Hazard Ratio, CI: confidence interval.

**Table 3 medicina-56-00282-t003:** Risk of stroke in relation to the commonly used CHM products.

The Commonly Prescribed CHMs	Ingredient of Generic Name	Function of CHMs	Frequency	Crude HR (95% CI)	Adjusted HR (95% CI) *
Fu Fang Dan Shen Pian (FFDSP)	Dan Shen (*Salvia miltiorrhiza Bunge*), San Qi (*Panax notoginseng (Burkill) F.H.Chen*) and Bing Pian (*Dryobalanops aromatica Gaertn. f.*)	Anti-atherosclerosis by dilating the cerebral vessels, inhibiting the aggregation of platelets, activating circulation, dispersing blood stasis	2563	0.78 (0.57–1.07)	0.76 (0.55–1.05)
Shu Jing Hwo Shiee Tang (SJHST)	Dang Gui (*Angelica sinensis* Oliv. Diels), Gan Cao (*Radix Liquiritiae Fisch*), Bai Shao (*Paeonia lactiflora Pallas*), Sheng Di Huang (*Radix Rehmanniae Recens*), Cang Zhu (*Atractylodes lancea (Thunb.) DC.*), Niu Xi (*Achyranthes bidentata Blumb*), Chen Pi (*Citrus reticulata Blanco*), Wei Ling Xian(*Clematis chinensis Osbeck.*), Fang; Ji (*Stephania tetrandra S. Moore*), Qiang Huo (Notopterygium incisum Ting ex H. T. Chang), Bai Zhi (*Angelicae Dahuricae Radix*), Long Dan Cao (*Gentiana scabra Bge.*), Chuan Xiong (*Ligusticum striatum DC*.), Tao Ren (*Prunus persica L. Batsch*), Fu Ling (*Poria cocos (Schw.) Wolf*) and Sheng Jiang (*Zingiber officinale Rosc.*)	Anti-coagulation effect, anti-hypersensitivity effects, increasing blood circulation and relieving pain	4518	0.67 (0.60–0.76)	0.66 (0.60–0.75)
Bu Yang Huan Wu Tang (BYHWT)	Huan Qi (*Astragalus membranaceus Fisch.*), Dang Gui Tail (*Angelica sinensis Oliv. Diels*), Chi Shao Yao (*Paeonia lactiflora Pall.* or *Paeonia veitchii* Lynch), Chuan Xiong (*Ligusticum striatum DC.*), Tao Ren (*Prunus persica L. Batsch*), Hong Hua (*Carthamus tinctorius* L.) and Di Long (*Pheretima aspergillum (Perrier)* or *Allolobophora caliginosa (Savigny) Trapezoides (Ant.* *Duges)*)	Anti-coagulation, removing blood stasis, neuroprotective and neurogenesis-promoting effects	2751	0.92 (0.71–1.12)	0.91 (0.72–1.15)
Tong Qiao Huo Xue Tang (TQHXT)	Chi Shao Yao (*Paeonia lactiflora Pall. or Paeonia veitchii Lynch*), Chuan Xiong (*Ligusticum striatum DC.*), Tao Ren (*Prunus persica* L. Batsch), Hong Hua (*Carthamus tinctorius* L.), She Xiang (*Moschus berezovskii Flerov*), Cong (*Alium fistulsum* L.), Sheng Jiang (*Zingiber officinale Rosc.*) and Da Zao (*Ziziphus jujuba Mill.*)	Removing blood stasis, neuroprotective effects by inhibiting inflammatory responses and apoptosis	4735	0.35 (0.12–0.84)	0.34 (0.13–0.82)
Xue Fu Zhu Yu Tang (XFZYT)	Chai Hu (*Radix Bupleuri*), Dang Gui (*Angelica sinensis Oliv. Diels*), Sheng Di Huang (*Radix Rehmanniae Recens*), Chi Shao Yao (*Paeonia lactiflora Pall. or Paeonia veitchii Lynch*), Hong Hua (*Carthamus tinctorius* L.), Tao Ren (*Prunus persica L. Batsch*), Zhi Qiao (*Fructus Aurantii Immaturus*), Gan Cao (*Radix Liquiritiae Fisch*), Chuan Xiong (*Ligusticum striatum DC.*), Niu Xi (*Achyranthes bidentata Blumb*), and Jie Geng (*Platycodon grandiflorum (Jaoq.) A.DC.*)	Removing blood stasis, neuroprotective effects by inhibiting inflammatory responses and apoptosis	6190	0.66 (0.57–0.76)	0.67 (0.58–0.77)
Ge Xia Zhu Yu Tang (GXZYT)	Dang Gui (*Angelica sinensis Oliv. Diels*), Chuan Xiong (*Ligusticum striatum DC.*), Chi Shao Yao (*Paeonia lactiflora Pall. or Paeonia veitchii Lynch*), Tao Ren (*Prunus persica L. Batsch*), Wu Ling Zhi (*Trogopterus xanthippes Milne-Edwards*), Hong Hua (*Carthamus tinctorius* L.), Mu Dan Pi (*Paeonia suffruticosa Andr.*) and Wu Yao (*Lindera aggregata (Sims) Kosterm.*), Yan Hu Suo (Corydalis yanhusuo W. T. Wang), Gan Cao (*Radix Liquiritiae Fisch*), Xiang Fu (*Cyperus rotundus Linn*) and Zhi Ke (*Citrus aurantium* L.)	Activating blood to dispel stasis, Invigorating qi movement and relieving pain	5273	0.59 (0.35–0.93)	0.58 (0.35–0.94)
Fu Yuan Huo Xue Tang (FYHXT)	Dang Gui (*Angelica sinensis Oliv.* *Diels*), Da Huang (*Rheum rhabarbarum* L.), Tao Ren (*Prunus persica L. Batsch*), and Chai Hu (*B.chinense DC. Or B.scorzonerifolium Willd*), Hong Hua (*Carthamus tinctorius* L.), Tian Hua Fen (*Trichosanthis Kirlowii Radix*) and Gan Cao (*Radix Liquiritiae Fisch*)	Invigorating blood circulation and dispelling blood stasis	6140	0.69 (0.51–0.92)	0.68 (0.50–0.92)
Tao He Cheng Qi Tang (THCQT)	Da Huang (*Rheum rhabarbarum* L.), Tao Ren (*Prunus persica L. Batsch*), Gui Zhi (*Ramulus Cinnamomi Cassiae*), Mang Xiao (*Natrii Sulfas*) and Gan Cao (*Radix Liquiritiae Fisch*)	Invigorating blood circulation and eliminating distension and spasmodic pain	4494	0.55 (0.35–0.86)	0.54 (0.35–0.85)
Tao Hong Si Wu Tang (THSWT)	Shu Di Huang (*Rehmannia glutinosa Liboschitz*), Bai Shao (*Paeonia lactiflora Pallas*), Dang Gui (*Angelica sinensis (Oliv.) Diels*), Chuan Xiong (*Ligusticum chuanxiong Hort.*), Tao Ren (*Prunus persica* (L.) Batsch.), and Hong Hua (*Carthamus tinctorius* L)	Promoting blood circulation and neuroprotective activity	7877	0.45 (0.31–0.67)	0.42 (0.30–0.63)
Shen Tong Zhu Yu Tang (STZYT)	Qin Jiao (*Radix Gentianae Macrophyllae*), Dang Gui (*Angelica sinensis Oliv. Diels*), Chuan Xiong (*Ligusticum striatum DC.*), Tao Ren (*Prunus persica L. Batsch*), Hong Hua (*Carthamus tinctorius* L.), Wu Ling Zhi (*Trogopterus xanthippes Milne-Edwards*), Niu Xi (*Achyranthes bidentata Blumb*), Qiang Huo (*Notopterygium incisum Ting ex H. T. Chang*), Mo Yao (*Commiphora myrrha Engl*), Xiang Fu (*Cyperus rotundus Linn*), Di Long (*Pheretima aspergillum (Perrier)* or *Allolobophora caliginosa (Savigny) Trapezoides (Ant.* *Duges)*) and Gan Cao (*Radix Liquiritiae Fisch*)	Removing blood stasis and anti-inflammatory effect	5934	0.68 (0.56–0.82)	0.66 (0.56–0.81)

* Model adjusted for age, sex, urbanization level, monthly income, and comorbidities.

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
