# Peer review of "Reduced Stroke Risk among Patients with Atrial Fibrillation Receiving Chinese Herbal Medicines Treatment: Analysis of Domestic Data in Taiwan"

_medicina, 2020, doi:10.3390/medicina56060282_

Round 1

Reviewer 1 Report

It is a nice paper with good research design. I have the following comments/ suggestions for your consideration.

  1. For diagnosis code of AF and stroke, how many diagnosis fields in the dataset. Is it primary diagnosis (first diagnosis), or any diagnosis filed (2nd, 3rd, or 4th diagnosis). Clarify this will be helpful for the readers.
  2. Index date of AF, for non-CHMs user, it is first diagnosis date. But for CHMs users, it is first date of prescriptions. We are not sure how many days after AF diagnosis the patients started took the CHMs, and this will be bias between the comparison groups. I would use the first AF diagnosis as index date for both group.
  3. I guess the starting treatment date is different between patients, using time-dependent variable to explore their associations will be helpful if the variable is available.  How to calculate the treatment period, how about the discontinued treatment, and then resumed, how about the medication switching, discuss these will improve the quality of the paper.
  4. For propensity score matching, I am not clear meaning from line 97-104? Why you choose "MENOPAUSAL WOMEN” as control group?

Author Response

Dear Editor-in-Chief

Thank you for your positive response and the constructive comments from the Reviewers. We followed closely the Reviewers’ comments and the Journal's guidelines for authors in revising our manuscript. In the revised manuscript, we highlighted all amendatory material with red color. Our point-by-point responses to the Reviewers’ concerns and suggestions are listed below. We hope that our detailed responses are satisfactory.

Reviewer 1

Q1: For diagnosis code of AF and stroke, how many diagnosis fields in the dataset. Is it primary diagnosis (first diagnosis), or any diagnosis filed (2nd, 3rd, or 4th diagnosis). Clarify this will be helpful for the readers.

Response: In the LHID, there are three diagnostic fields in the outpatient service claims, and four diagnostic fields in the inpatient claims. As to the definition of AF in this work, we not only applied the commonly used approach, namely including at least 3 outpatient service claims or being hospitalized with at least one principal diagnosis[1, 2], but also considered the prescription pattern to markedly increase the validity of AF diagnosis. Please refer to lines 23-25 on Page 5. With regard to the stroke incident, it was identified by the catastrophic illness registry. Only those with catastrophic illness certification due to stroke (ICD-9-CM code 430-438) were recruited. Please refer to lines 1-5 on Page 6.

Q2: Index date of AF, for non-CHMs user, it is first diagnosis date. But for CHMs users, it is first date of prescriptions. We are not sure how many days after AF diagnosis the patients started took the CHMs, and this will be bias between the comparison groups. I would use the first AF diagnosis as index date for both group.

Response: After converting the initial index date into the first AF diagnosis date for both group, we found that the results from the re-analysis indicated that adding CHMs to convention therapy was still related to decreased risk of stroke, with the adjusted HR of 0.72 (95% CI: 0.65-0.79).

Q3: I guess the starting treatment date is different between patients, using time-dependent variable to explore their associations will be helpful if the variable is available.  How to calculate the treatment period, how about the discontinued treatment, and then resumed, how about the medication switching, discuss these will improve the quality of the paper.

Response: In our study, for the CHMs user group, the time period between the date of AF diagnosis and the date of first CHMs treatment after AF onset represented the immortal time, which would tend to overestimate the intervention’s beneficial effect. We, therefore, assigned the first date of the initiation of CHMs treatment as the index date of the follow-up period for AF cases with CHMs use. This approach is often regarded as an alternative solution to account for the immortal time bias in cohort studies [2-4]. Please refer to the lines 16-19 of Page 6. The treatment period of CHMs was summed up based on the prescription days of herbal formulae by Chinese medicine physicians. Please refer to the lines 7-9 on Page 6.

Q4: For propensity score matching, I am not clear meaning from line 97-104? Why you choose "MENOPAUSAL WOMEN” as control group?

Response: This omission has been amended. Please refer to lines 11-14 on Page 6.

References

  1. Lu, M.C., et al., Bidirectional associations between rheumatoid arthritis and depression: a nationwide longitudinal study. Scientific Reports, 2016. 6: p. 20647.
  2. Tsai, T.Y., et al., Decreased risk of stroke in patients receiving traditional Chinese medicine for vertigo: a population-based cohort study. J Ethnopharmacol, 2016. 184: p. 138-143.
  3. Lévesque, L.E., et al., Problem of immortal time bias in cohort studies: example using statins for preventing progression of diabetes. BMJ, 2010. 340.
  4. Wu, M.Y., et al., Acupuncture decreased the risk of coronary heart disease in patients with fibromyalgia in Taiwan: a nationwide matched cohort study. Arthritis Research & Therapy, 2017. 19(1): p. 37.
  5. Camm, A.J., et al., Guidelines for the management of atrial fibrillation: the task force for the management of atrial fibrillation of the European society of cardiology (ESC). European Heart Journal, 2010. 31(19): p. 2369-2429.
  6. Taiwan_National_Adverse_Drug_Reactions_Reporting_System. Taiwan_National_Adverse_Drug_Reactions_Reporting_System. 2020 [cited 2020 01/05]; Available from: https://dep.mohw.gov.tw/DOCMAP/cp-3925-40834-108.html.

Reviewer 2 Report

In technical documents such as this, difficulty of comprehension arises from less than optimal use of English and I struggled to understand the import of a point being made in several places and had to re-read those sections.

Statistically, you have employed techniques to reduce classification and selection biases and the impact of confounding factors, and have acknowledged limits to these measures. Equally, you have differentiated those CHM products (FFDSP & BYHWT) the adjusted hazard ratios of which may be ≥1, according to their CIs. In your discussion section, should you not differentiate further the remaining CHM products quantitatively, according to their CIs? 

Author Response

Dear Editor-in-Chief

Thank you for your positive response and the constructive comments from the Reviewers. We followed closely the Reviewers’ comments and the Journal's guidelines for authors in revising our manuscript. In the revised manuscript, we highlighted all amendatory material with red color. Our point-by-point responses to the Reviewers’ concerns and suggestions are listed below. We hope that our detailed responses are satisfactory.

Reviewer 2

Q1: In technical documents such as this, difficulty of comprehension arises from less than optimal use of English and I struggled to understand the import of a point being made in several places and had to re-read those sections.

 Statistically, you have employed techniques to reduce classification and selection biases and the impact of confounding factors, and have acknowledged limits to these measures. Equally, you have differentiated those CHM products (FFDSP & BYHWT) the adjusted hazard ratios of which may be ≥1, according to their CIs. In your discussion section, should you not differentiate further the remaining CHM products quantitatively, according to their CIs?

Response: We invited a professional editor (Professor Hanoch Livneh) who is a fluent English speaker, and serves on the editorial boards of several English-speaking psychological and medically-related journals. He is also well acquainted with the American Medical Writers Association guidelines. Dr. Livneh assisted in editing the content, spelling and grammar of this manuscript.

  Also we provided additional explanations to account for the insignificant impacts of FFDSP & BYHWT. Please refer to lines 5-9 on Page 10, We also indicated that caution should be given when interpreting the findings, especially regarding the therapeutic efficacy of herbal formulae. We anticipate that this pilot study could be regarded as a basis for further pharmacological investigations and clinical trials that overcome the limitations of this study. Accordingly, we rewrote part of the discussion to clarify our findings. Please refer to lines 20-26 on Page 10, as well as lines 8-11 on Page 11.

References

  1. Lu, M.C., et al., Bidirectional associations between rheumatoid arthritis and depression: a nationwide longitudinal study. Scientific Reports, 2016. 6: p. 20647.
  2. Tsai, T.Y., et al., Decreased risk of stroke in patients receiving traditional Chinese medicine for vertigo: a population-based cohort study. J Ethnopharmacol, 2016. 184: p. 138-143.
  3. Lévesque, L.E., et al., Problem of immortal time bias in cohort studies: example using statins for preventing progression of diabetes. BMJ, 2010. 340.
  4. Wu, M.Y., et al., Acupuncture decreased the risk of coronary heart disease in patients with fibromyalgia in Taiwan: a nationwide matched cohort study. Arthritis Research & Therapy, 2017. 19(1): p. 37.
  5. Camm, A.J., et al., Guidelines for the management of atrial fibrillation: the task force for the management of atrial fibrillation of the European society of cardiology (ESC). European Heart Journal, 2010. 31(19): p. 2369-2429.
  6. Taiwan_National_Adverse_Drug_Reactions_Reporting_System. Taiwan_National_Adverse_Drug_Reactions_Reporting_System. 2020 [cited 2020 01/05]; Available from: https://dep.mohw.gov.tw/DOCMAP/cp-3925-40834-108.html.

Reviewer 3 Report

retrospective observational propensity score study.

Surprisingly high stroke rate, given the relatively young age (under 60) and ½ were male – the control group has CVA rate of CHAâ‚‚DSâ‚‚-VASc Score of 4, very atypical of representative AF cohort. Please explain.

The reviewer requests the number of patients in each class of CHAâ‚‚DSâ‚‚-VASc Score be compared between the study and propensity matched populations.

Please show what other drugs these patients are on, most AF patients will be on other "western" meds and potential drug interactions need to be shown and addressed. 

Its highly unlikely the study is powered to look at individual CHM (10 listed), furthermore no analysis of the frequency, combination, and outcome of those taking two of more of CHMS listed. Please address.

The hypothesis of antiplatelet effect of CHMs providing incremental CVA prevention over Warfarin is probably quite a weak one at best, given the strong clinical data of aspirin failing to do so, in fact concomitant use of aspirin and warfarin consistently showed increased signal of excess bleed including intracranial bleed and hemorrhagic CVA.   A more credible discussion of potential mechanisms is recommended. 

For each CHM, if any is, powered to predict complementary effect to reduce CVA, the most likely pharmacotherapeutic component should be elaborated.

The conclusion statements “The present study reported that adding CHMs to conventional therapy may decrease the sequent stroke risk for AF subjects.” and “Results of this study may   serve as a reference for healthcare providers in helping to establish more effective therapeutic   interventions to improve the prognosis for AF patients.” are nebulous and clinically meaningless if not reckless recommendation given CHMs represent hundreds if not thousands of compounds and mixture permutations.  Please revise.

Author Response

Dear Editor-in-Chief

Thank you for your positive response and the constructive comments from the Reviewers. We followed closely the Reviewers’ comments and the Journal's guidelines for authors in revising our manuscript. In the revised manuscript, we highlighted all amendatory material with red color. Our point-by-point responses to the Reviewers’ concerns and suggestions are listed below. We hope that our detailed responses are satisfactory.

Reviewer 3

Q1: Surprisingly high stroke rate, given the relatively young age (under 60) and ½ were male – the control group has CVA rate of CHAâ‚‚DSâ‚‚-VASc Score of 4, very atypical of representative AF cohort. Please explain.

Response: The incidence density of stroke was calculated as the number of cases per 1000 person-years (PYs). It was highlighted in the sections of the "Abstract", and the "Statistical analysis". Please refer to line 12 of Page 3, and lines 12-13 on Page 7.

  Additionally, we applied the propensity score approach to reduce the imbalance between two selected groups. For each AF patient who received CHMs, one control patient was selected by 1:1 matching, based on a propensity score. The propensity score was calculated using logistic regression on the basis of patients’ demographics and baseline comorbidities at enrollment (shown in Table 1). To further address the Reviewer's concern, we utilized the CHAâ‚‚DSâ‚‚-VASc Score to replace the original comorbidities indexes. The CHAâ‚‚DSâ‚‚-VASc Score was calculated based on the following algorithm containing eight variables [5], such as congestive heart failure, hypertension, diabetes, age≧75 years, prior ischemic/transient ischemic attack, vascular disease, age 65-74 years, and female. In response to the study purpose, we only recruited new-onset AF patients without any history of stroke to determine whether the use of CHMs would reduce the subsequent risk of stroke. To avoid double-counting and possible over-adjustment in the regression model, the factor (prior ischemic/transient ischemic attack) was excluded as a determinant of the CHAâ‚‚DSâ‚‚-VASc score in the following sensitivity analysis.

  After performing a sensitivity analysis, we noted that the CHAâ‚‚DSâ‚‚-VASc score between CHMs users and non-CHMs users was 1.87(±1.33) and 1.87(±1.38), respectively. The difference of CHAâ‚‚DSâ‚‚-VASc score between the two groups did not reach statistical significance (t= -0.13; p= 0.89). Thereafter, the CHAâ‚‚DSâ‚‚-VASc score was added while calculating the propensity score and further entered into the multivariate Cox regression. The beneficial effect of CHMs in reducing the development of stroke was still observed, with the adjusted HR of 0.70 (95% CI: 0.64-0.77). We believe that these additional data clarify the issue and address the Reviewer's concern.

Q2: The reviewer requests the number of patients in each class of CHAâ‚‚DSâ‚‚-VASc Score be compared between the study and propensity matched populations.

Response: To address this potential problem, we calculated, for each study patient, the propensity score based on the variables shown in Table 1, as well as the CHAâ‚‚DSâ‚‚-VASc score. Afterwards, we included the propensity score in the multivariate Cox regression to adjust for such potential confounding by indication. The estimated HR associated with CHMs was 0.70 (95% CI: 0.64-0.77), supporting that CHAâ‚‚DSâ‚‚-VASc score did not affect the findings of this study.

Q3: Please show what other drugs these patients are on, most AF patients will be on other "western" meds and potential drug interactions need to be shown and addressed.

Response: Following meticulous review of the Taiwan National Adverse Drug Reactions Reporting System [6], we found that there were no severe drug interaction effects occurring among Taiwanese AF patients who adopted the concomitant treatments of Western medicines and CHMs. However, owing to different living environments, physical conditions and geographic regions, the study findings may not directly apply to other countries. We acknowledge the validity of the Reviewer's concern, and we attempted to clear up this point in our discussion. Please refer to lines 20-26 of Page 10.

Q4: Its highly unlikely the study is powered to look at individual CHM (10 listed), furthermore no analysis of the frequency, combination, and outcome of those taking two of more of CHMS listed. Please address.

Response: We added the ingredients and use frequency of the herbal products based on the Reviewer's suggestion. Please refer to the revised table 3 in the revised manuscript. In addition to the ingredients and frequency of all Chinese herbal products, table 3 further shows the therapeutic effect that each herbal formula had on stroke onset. As we discussed, it is noteworthy that the enrolled subjects were not initially randomly categorized into users and nonusers, and the sample was limited to a single country as well. Caution, therefore, should be exerted when interpreting the findings, especially the assessment of Chinese herbal products efficacy. Given the possible beneficial effect of CHMs on subsequent stroke risk [2], and since only limited information existed on whether CHMs successfully modified the relationship between AF and stroke risk, we have argued that the preliminary findings of this study could be used as a basis for further pharmacological investigations and clinical trials. This issue has been noted in the “Limitations” section of the manuscript. Please refer to lines 20-26 of Page 10.

Q5: The hypothesis of antiplatelet effect of CHMs providing incremental CVA prevention over Warfarin is probably quite a weak one at best, given the strong clinical data of aspirin failing to do so, in fact concomitant use of aspirin and warfarin consistently showed increased signal of excess bleed including intracranial bleed and hemorrhagic CVA. A more credible discussion of potential mechanisms is recommended.

Response: We appreciate the reviewer's comment. The main purpose of this study was to determine whether the integration of CHMs into conventional medicine can lower the subsequent risk of stroke onset, and not to directly compare the effects of CHMs and Warfarin as mentioned by the Reviewer. Factually, in our paper, AF was defined by a compatible ICD-9-CM code, plus the prescription of Warfarin for at least three months. (Please refer to lines 23-25 on Page 5). It is, therefore, conceivable that the Reviewer may have misinterpreted the purpose of our study. Although the findings of our study provided initial support to the argument that integrative CHMs could reduce stroke risk for AF patients in Taiwan, further clinical and mechanistic studies are still warranted to strengthen these findings. We acknowledged these points in the “Limitations” section of the manuscript. Please refer to lines 20-26 of Page 10, and lines 8-11 on Page 11.

Q6: For each CHM, if any is, powered to predict complementary effect to reduce CVA, the most likely pharmacotherapeutic component should be elaborated.

Response: As suggested by the Reviewer, the ingredients of each Chinese herbal product have been individually illustrated in the revised table 3.

Q7: The conclusion statements “The present study reported that adding CHMs to conventional therapy may decrease the sequent stroke risk for AF subjects.” and “Results of this study may  serve as a reference for healthcare providers in helping to establish more effective therapeutic  interventions to improve the prognosis for AF patients.” are nebulous and clinically meaningless if not reckless recommendation given CHMs represent hundreds if not thousands of compounds and mixture permutations. Please revise.

Response: As indicated by the Reviewer, the relevant statement has been rephrased. Please refer to lines 8-11 on Page 11.

References

  1. Lu, M.C., et al., Bidirectional associations between rheumatoid arthritis and depression: a nationwide longitudinal study. Scientific Reports, 2016. 6: p. 20647.
  2. Tsai, T.Y., et al., Decreased risk of stroke in patients receiving traditional Chinese medicine for vertigo: a population-based cohort study. J Ethnopharmacol, 2016. 184: p. 138-143.
  3. Lévesque, L.E., et al., Problem of immortal time bias in cohort studies: example using statins for preventing progression of diabetes. BMJ, 2010. 340.
  4. Wu, M.Y., et al., Acupuncture decreased the risk of coronary heart disease in patients with fibromyalgia in Taiwan: a nationwide matched cohort study. Arthritis Research & Therapy, 2017. 19(1): p. 37.
  5. Camm, A.J., et al., Guidelines for the management of atrial fibrillation: the task force for the management of atrial fibrillation of the European society of cardiology (ESC). European Heart Journal, 2010. 31(19): p. 2369-2429.
  6. Taiwan_National_Adverse_Drug_Reactions_Reporting_System. Taiwan_National_Adverse_Drug_Reactions_Reporting_System. 2020 [cited 2020 01/05]; Available from: https://dep.mohw.gov.tw/DOCMAP/cp-3925-40834-108.html.

Round 2

Reviewer 3 Report

the authors should consider adding statements in abstract, conclusion as well as in discussion, to the effect that the study is not powered to determine stroke prevention properties of CHMs in AF (due to heterogeneity and multiplicity of CHMs subscribed by the users). The study findings are at present not clinically meaningful and useful, pending results of future, well powered RCTs.

Author Response

Dear Editor-in-Chief

Thank you for your positive response and the constructive comments. We still followed closely the Reviewers’ comments and the Journal's guidelines for authors in revising our manuscript. In the revised main manuscript, we highlighted all amendatory material with red color. Our point-by-point responses to the Reviewers’ concerns and suggestions are listed below. We hope that our detailed responses are satisfactory.

Reviewer 3

Q1:  The authors should consider adding statements in abstract, conclusion as well as in discussion, to the effect that the study is not powered to determine stroke prevention properties of CHMs in AF (due to heterogeneity and multiplicity of CHMs subscribed by the users). The study findings are at present not clinically meaningful and useful, pending results of future, well powered RCTs.

Response: As suggested by the Reviewer, we added some suggestions to address the need for caution when interpreting the findings of this study. We also agree that the mechanisms underlying CHMs use on stroke susceptibility warrant further investigation via randomized control trials. Please refer to lines 13-17 on Page 3, lines 24-26 on Page 10, as well as lines 9-12 on Page 11.
